# Factors Associated with Behavioral Disorders in Children with Congenital Zika Syndrome and Their Families—A Cross-Sectional Study

**DOI:** 10.3390/ijerph19159554

**Published:** 2022-08-03

**Authors:** Nívia Maria Rodrigues Arrais, Claudia Rodrigues Souza Maia, Nathália Allana de Amorim Rodrigues, Rafaela Silva Moreira, Valeria Azevedo de Almeida, Silvana Alves Pereira, Maria Isabel de Moraes Pinto

**Affiliations:** 1Pediatrics Department, Universidade Federal do Rio Grande do Norte, Natal 59077-010, Brazil; claudiasouzamaia@gmail.com; 2Pediatrics Department, Universidade Federal de São Paulo—UNIFESP, Sao Paulo 04021-001, Brazil; m.isabelmp@gmail.com; 3Physical Therapy Department, Universidade Federal do Rio Grande do Norte, Natal 59077-010, Brazil; nathalia.rodrigues.107@ufrn.edu.br (N.A.d.A.R.); valeria@edu.isd.org.br (V.A.d.A.); 4Department of Health Sciences, Universidade Federal de Santa Catarina, Ararangua 88905-120, Brazil; rafaela.moreira@ufsc.br

**Keywords:** Zika virus, child development, behavior, psychosocial risk, survey of well-being of young children

## Abstract

The Zika virus was responsible for an outbreak between 2015 and 2016 in Brazil: an alarming public health problem of international relevance. The Congenital Zika Syndrome (CZS) is often associated with manifestations that are responsible for cognitive and motor development delays and behavioral disorders. Thus, we aimed to characterize the clinical-epidemiological and familial context of those children and to identify factors associated with the risk of behavioral disorders using the Survey of Well-Being of Young Children questionnaire (SWYC). In total, 52 children diagnosed with CZS were evaluated. Logistic regressions were employed to assess predictive variables for behavioral alteration. Eighteen (35%) of the children presented a risk of behavioral alteration. Children born normocephalic were 36-fold more likely to present behavioral alteration (95% CI: 3.82 to 337.92, *p* = 0.002). Children with hearing and visual impairments showed reduced risks. In total, 35% percent of families reported food insecurity and 21% were at risk for maternal depression. Our findings suggest better social interactions and conditions to externalize reactions for children with CZS born normocephalic. The continuous assessment of these children and families may identify conditions associated with behavioral alteration and psychosocial vulnerabilities that help in decision-making, therefore optimizing patient–family interactions.

## 1. Introduction

The Zika virus (ZIKV) was responsible for an outbreak between 2015 and 2016 in Brazil: an alarming public health problem of international relevance [1,2]. It is a teratogenic arthropod-borne flavivirus transmitted from mother to fetus that can manifest in the neonate with a pattern of signs and symptoms recognized as Congenital Zika Syndrome (CZS) [3]. Since 2015, the Health Ministry has confirmed more than 4395 confirmed or probable cases of CZS in Brazil [4].

CZS is often associated with a global developmental delay [5]. Children with CZS may have different clinical manifestations such as microcephaly, muscle and joint contractures, epilepsy, central and peripheral nervous system lesions, as well as hearing, visual, and intellectual deficits [6,7]. All those manifestations are responsible for cognitive and motor development delays and behavioral disorders [1,6,7]. Regarding the behavioral disorders, studies have described irritability from birth [7,8], while others have observed children with autism spectrum disorder (ASD) after exposure to Zika virus infection during pregnancy [8,9]; however, the background investigations of factors that may increase the predisposition to behavioral disorders in children with CZS is very restricted [3]. Epidemiological studies have shown that most pregnant women exposed to ZIKV were from northeastern Brazil [1], with low per capita income and very poor housing and living conditions [10]. Cunha et al. [5] showed that the incidence rate of ZIKV infection was associated with physical, psychological, and moral violence and homicides in northeastern Brazil. This scenario draws attention to the contextual vulnerability to which children diagnosed with the syndrome are inserted and highlights the importance of analyzing the possible impacts of this reality in the long term.

Furthermore, unfavorable socioeconomic conditions and deprivation of the family environment negatively impact the development of children with disabilities [1,11]. These adversities are often severe barriers to adequate access to public health and the specific multidisciplinary follow-up required by children with neurological impairments [11]. Family conditions such as maternal depression, domestic violence, and socioeconomic factors may directly affect the binomial mother-child and make children vulnerable to behavioral disorders [11,12,13]. Despite the increasing understanding of morphological alterations of fetal ZIKV infection and its developmental impairments [1,7], the long-term behavior implications of congenital ZIKV exposure and the interrelationship with unfavorable socioeconomic and family conditions remain poorly investigated. Such an approach may help to detect skills that were expected but not achieved and the children at risk of behavior disorder profiles. Earlier and more targeted multidisciplinary interventions may address the peculiar needs of children with CZS. In this study, we aimed to characterize the epidemiological, clinical, and familial context of children with CZS, and to identify factors associated with the risk of behavioral disorders using the Survey of Well-Being of Young Children questionnaire (SWYC).

## 2. Materials and Methods

This is cross-sectional study recruited children with CZS from the outpatient clinic for congenital infections of a university hospital in northeastern Brazil where 71 children with a diagnosis of CZS according to the Brazilian Ministry of Health (2017) [2] were regularly followed-up between November 2015 and June 2020.

This study included 52 children whose parents responded to the SWYC questionnaire and routine clinical assessment. It was approved by the research ethics committee of Universidade Federal do Rio Grande do Norte (CAAE:57444016.1.0000.5292) and Universidade Federal de São Paulo/Escola Paulista de Medicina (CAAE:57444016.1.3001.5505). All guardians signed an informed consent form.

### 2.1. Study Procedures

The socioeconomic classification was evaluated according to Associação Brasileira de Empresas de Pesquisa—ABEP [14] during the medical appointments. The stratification was made considering the purchasing power of items listed in the instrument, and the educational level of the family provider. Maternal data (age at child delivery; educational level; presence of symptoms of Zika virus infection during pregnancy, such as fever, rash, or pruritus; and the trimester when the maternal symptoms appeared). Children’s data (gender, gestational age at birth, head circumference, weight and height at birth and at 36 months, exclusive breastfeeding up to 6 months, convulsion, arthrogryposis, gastrostomy need, fundoscopy and auditory acuity testing) were obtained in follow-up visits at the clinic and compiled in an Excel^®^ spreadsheet.

Prematurity was defined as delivery before 37 weeks of gestational age. Microcephaly was defined when the head circumference was less than two standard deviations below the mean for gestational age and sex. Normocephaly was defined when the head circumference was above or equal to two standard deviations above the mean [2,15]. The presence or absence of spastic tetraparesis and seizures was confirmed by a neurologist who was responsible for monitoring the children. Arthrogryposis was defined as joint deformity resulting from central neurological impairment [7].

### 2.2. Data Collection

To assess the socioemotional factors and family context, guardians answered the SWYC questionnaire individually in person or by telephone in a 15 min interview carried out between November 2019 and April 2020 [16,17]. Although there are several instruments available for identifying signs of risk for developmental delays, the SWYC is particularly advantageous due to its ease of use as a first-line screening tool and its ability to assess children with varying degrees of neurological impairment [3]. The following age-specific domains were applied in the interview: (1) ten questions on developmental milestones; (2) eighteen questions on preschool pediatric symptom checklist (PPSC) to investigate externalizing (e.g., Does your child have trouble playing with other children? Is your child aggressive?) and internalizing behaviors (e.g., Does your child seem nervous or afraid? Does your child seem sad or unhappy?); (3) two questions on parents’ concern regarding development and behavior; and (4) ten family questions on tobacco, alcohol and drug use, and risk of food insecurity, maternal depression, and conflicts between parents [16,17]. Only preschool-age children were included in an attempt to offer greater homogeneity in relation to clinical observation.

The classification of altered developmental milestones followed age-specific scores described in the SWYC-BR manual [17]. In the PPSC, the risk for behavioral disorders was considered when the total score was ≥9. The section assessing parental concern on child behavior and learning and development was qualitatively classified according to Perrin et al. (2016) [16] and the SWYC-BR manual [17]. In the family questions domain, risk for tobacco, alcohol, or drug use and food insecurity were considered when the guardian answered “yes” to any of the questions within this domain; depression, when the total scores referring to this item was ≥3; and risk for domestic violence, when the guardian answered “A lot of tension” and/or “Great difficulty” for at least one of two questions in this domain [16,17].

In order to assess the developmental milestones for the age in more detail, a Gross Motor Function Classification System [GMFCS] was evaluated during the interview. Maternal information and/or clinical assessments were also assessed considering different aspects such as: does the child pick up objects, sit up alone, and express him/herself with a word?

### 2.3. Statistical Analysis

Categorical and numerical variables were presented as relative (%) and absolute frequencies using SPSS-IBM SPSS, (v.20.0, Armonk, NY, USA) and STATA 17 (StataCorp. 2021. Stata Statistical Software: Release 17. StataCorp LLC.: College Station, TX, USA). An unpaired test was used to compare parametric data. Univariate and multivariate logistic regressions or Firth regression models were adjusted according to sample size to assess variables predicting the risk of behavioral disorders. In the univariate regression, variables with significant association at 10% were selected, whereas non-significant variables at 5% were excluded using the backward method, except the age of the child, which was kept as a control variable, regardless of its significance.

## 3. Results

Fifty-two out of the seventy-one children diagnosed with CZS who were regularly followed-up at the outpatient clinic were included in the study. Out of the 19 children excluded, 7 died before the interview, and 12 did not attend the last scheduled appointment. Mean age at assessment was 50 months (range, 22 to 61 months), 28 (54%) were male, and 41 (79%) were born with gestational age greater than 37 weeks. Incomplete high school education prevailed in 48% of mothers, and 61% of families were classified in socioeconomic stratum C (Table 1).

All children assessed presented altered central nervous system imaging exams and had spastic tetraparesis. Forty-seven children (90%) were classified as GMFCS V, seven (14%) had arthrogryposis, and five (10%) had a gastrostomy. Fifteen children (29%) were able to speak at least one word, and only nine (17%) spoke several words and some complete sentences (Table 2).

Developmental milestones were altered in 100% of children; 35% of children were at risk for behavioral disorders according to PPSC; 42 (81%) mothers were concerned about the child’s development and 24 (46%), about his/her behavior; 31% of families were at risk for tobacco, alcohol, or drug abuse, and 35% were at risk for food insecurity; 21% of mothers were at risk for depression; and 8% for domestic violence (Table 2).

The risk of behavioral disorders was associated with motor ability to pick up objects (*p* = 0.002) and to sit up alone (*p* = 0.005). Non-significant associations were observed between behavioral disorders and verbal communication (*p* = 0.071), food insecurity (*p* = 0.451), and risk for maternal depression (*p* = 0.159). Table 3 presents the association between behavioral disorders and family and child variables.

At 2 years of age, the altered auditory exam decreased in 92%, and strabismus decreased, and so did 95% of the odds of risk of behavioral disorders. By contrast, normocephalic children had higher odds of being at risk of behavioral disorders than microcephalic children. Table 4 presents the final logistic regression model for the risk of behavioral disorders.

## 4. Discussion

This study showed that 35% of children diagnosed with CZS were at risk for behavioral disorders. Those born normocephalic were 36-fold more likely to display behavioral disorders than those born with microcephaly; by contrast, children with altered auditory evaluation and strabismus presented less risk of behavioral disorders than those without these alterations. Microcephaly and developmental delay were present in the first year of life, as observed by other groups [3,6,18]. Earlier and more severe central nervous system lesions may explain the differences in development and behavior between those born microcephalic and those born normocephalic, and who subsequently evolved to microcephaly. Aragão et al. [19] evaluated computed tomography scans of the skull of children with CSZ born normocephalic and reported a less severe impairment when compared with those born microcephalic. The former showed polymicrogyria and the latter displayed severe cerebral atrophy, pachygyria, and lissencephaly. Our data may indicate that children born normocephalic seem to have better conditions of interacting and reacting to daily situations when compared to those born microcephalic, who would not display behavioral disorders. Despite the fact that both groups had intense neurological impairment at the time of application of the SWYC questionnaire, the difference in the development of microcephaly might be associated with different abilities to externalize behavioral disorders.

The higher chances of behavioral disorder observed in children born normocephalic are similar to Sobral et al.’s findings [3]. Using the SWYC-BR instrument to investigate patterns of neurodevelopment and behavior in groups of children with different degrees of ZIKV, they observed that greater levels of neuro-psychomotor impairment lead to the simplification or omission of behavioral reactions, which may compromise the assessment proposed by the instrument and result in an underestimated finding.

All children presented severe central nervous system damage, suggesting severe developmental impairment [7,10]. The proportion of 90% of the cohort classified as grade V of the GMFCS (e.g., the highest degree of motor impairment) is similar to the results reported by Takahasi et al. (89%) [20] and Melo et al. (81%) [21] and reflects a poorer functional ability and less independency to perform motor tasks, probably due to the severe brain damage associated with CZS. Furthermore, in the study by Takahasi et al. [20], the CZS children classified as GMFCS level V tended to reach their maximal gross motor function potential relatively early, by their second birthday, and to underperform on the analysis of gross motor function when compared with cerebral palsy children with the same GMFCS classification [20]. This probably happens because of the severe brain damage associated with CZS and the progressive impairment inherent to the syndrome [6]. It is worth to note that seven (10%) children died in the first two years of life, similar to data in the literature [22], reflecting the severe condition of children with CSZ.

Hearing impairment in children may hinder communication and interfere with social interactions [6,23]. In our study, we demonstrated that children with altered audiological and visual exams were less likely to present behavioral modification. The low risk of behavioral disorders observed in children born with microcephaly and those with hearing and visual impairments may reflect low exposure to conflicting situations or limitation of negative interactions due to severe neurological impairment [3].

Regarding the family context, approximately one third of the families were at risk for tobacco, alcohol, or drug abuse and for food insecurity. In the maternal context, 21% of the mothers were at risk for depression, 48% had not completed high school, and 61% belonged to socioeconomic class C. Children living in adverse environments such as parenting drug abuse and a low socioeconomic class are more likely to present developmental delay, difficulties to access health services, and precarious survival conditions [18,24]. Of interest, previous literature reveals that smoking and illicit drug use during pregnancy are risk factors for the development of microcephaly in children born to mothers exposed to ZIKV in the perinatal period [25].

Studies have already detected changes in the family dynamics of children with CZS, especially in maternal dynamics. Mothers tend to significantly reduce their participation in formal work and to interrupt their education due to the almost exclusive dedication to the care of the child, which, consequently, decreases the maternal quality of life and increases the risk of food insecurity [10,26]. It should be noted that previous literature showed that nutritional deficiency in important periods of intra- and extra-uterine development, as observed in food insecure families, could cause irreversible damage to intellectual potential and behavior [25], and be a risk factor for cognitive and socioemotional developmental delays [27]. Although we do not have data on gestational nutrition, the finding of a high proportion of risk for food insecurity at the time of the interview should be a warning to start interventions and support for families.

Studies that have also investigated children with CZS reported a similar pattern of the maternal educational and socioeconomic profile [20,25]. A proportion of 52% [25] and 63% [20] of the mothers had incomplete or complete high school education. Regarding the family socioeconomic profile, a proportion of 51% [25] and 46% [20] of the families occupied a low socioeconomic level.

The mental health of parents is interlinked with the health conditions of children [28]. The proportion of risk for maternal depression found in our sample is similar to those from other studies [22,28,29]. Laza-Vásquez et al. [26] identified situations associated with stress and sadness in reports of mothers of children with CZS, such as anguish, fear of the future, permanent alertness about child health, concern about resources, and reduced self-care and social life. Interestingly, the risk of behavioral disorders was not associated with maternal depression. Studies with larger samples and qualitative details are needed to clarify the caregiver–child interaction.

Questions regarding behavior for the age bracket of the sample group did not correspond to cognitive, language, and motor skills of the children, and that may have influenced assessments. Other instruments can contribute to a more detailed evaluation, such as the Bayley III scale that reports global impairment [8]. The SWYC questionnaire, cross-culturally adapted to Brazilian Portuguese by Moreira et al. (2019) [8], is easily applicable in daily practice and was rarely used in children with CZS. This instrument uses a list of questions related to internalizing and externalizing the behaviors of children, addressing difficulties in socio-adaptive routines. Results of the SWYC depend on child exposure to stressful people, environmental situations, and the possibility of reactions. Although the adequacy of some questions for children with severe neurological limitations may lead to answers that do not represent their reality [3], the SWYC questionnaire is an excellent tool for screening behavioral disorders.

The spectrum of clinical manifestations in CZS is broad [2,23]. The SWYC questionnaire can help identify risks of behavioral disorders and altered family contexts in children with severe motor impairment. Despite that, this instrument could be used to monitor early behavioral disorders and enable early interventions in children exposed to ZIKV during pregnancy regardless of the clinical and neurological aspects noted at birth.

Considering the family context, caring for children with global developmental delays is challenging [6,10,28]. Unfavorable family environments are associated with adverse behavioral outcomes [9,27], mainly when children present with developmental delays. Although 35% of families reported food insecurity in the previous year, and 8% reported signs of family conflicts, associations were not observed with the risk of behavioral disorders.

The relatively small sample size and severe neurological impairment observed in most children can be considered study limitations. However, studies carried out with children affected by the Zika virus in Brazil have a sample size similar to that of the present study [1,6]. Freitas et al. [6] indicated that despite the fact that 59% of the published studies on Zika were carried out in Brazil, the samples of the studies are small. In this review, with 46 studies on Zika, 27 were carried out in Brazil and 22 had a sample number between 1 and 50 individuals. In this perspective, despite all the implications on the inference of the results, as well as the expressive values of confidence intervals of the OR, we understand that the final model can be useful in the discussion about the possible predictors of behavioral disorders in children with the Zika virus. Additionally, if we chose to perform a sample size calculation, we could run the risk of finding even lower values than those presented in the study. Although interviews were standardized, questions regarding domestic violence and tobacco, alcohol, and drug use may have been underestimated due to the questionnaire format.

This study characterized the family context of Brazilian children with CZS, and elucidated factors considered to be a risk for socioemotional behavioral disorders and psychosocial vulnerability. Although the study was performed in a region of Brazil with significant social and economic challenges, we used an easy-to-apply and accessible instrument that allows continuous monitoring of behavioral and family issues, helps in decision-making and in support of families, and optimizes resources. For this reason, in order to minimize risks of cross-cultural biases and to achieve reliable and comparable measures of the developmental and behavioral domains, this study utilized a validated translation of the SWYC.

## 5. Conclusions

Our results suggest that children with CZS born normocephalic presented a high risk of behavioral disorders, suggesting increased social interaction and good enough conditions that allow externalizing reactions. Those born with microcephaly and hearing and visual alterations presented a low risk of behavioral disorders, indicating limitations in socio-adaptive interactions and severe neurological impairment. To confirm these findings, we suggest that these children should be evaluated with repeated assessments using more accurate and comprehensive instruments, such as Bayley-III. It is worth mentioning that previous studies have already reported autistic spectrum disorder in these children. [7,30]. The SWYC may be adopted as a screening tool for behavioral disorders, and to assess situations of family vulnerability such as food insecurity and maternal depression, as found in this study. This will allow these children and their families to receive the necessary multidisciplinary care and support as early as possible.

## Figures and Tables

**Table 1 ijerph-19-09554-t001:** Maternal data distribution of children (*n* = 52) according to risk of behavioral alteration assessed with the Preschool Pediatric Symptom Checklist of the SWYC questionnaire.

Preschool Pediatric Symptom Checklist to Behavior Disorders						
Variables	At Risk		No Risk		Total	
	*n*	%	*n*	%	*n*	%
Maternal age at child birth	18	100	34	100	52	100
≤18 years	4	22	3	9	7	14
19 to 35 years	9	50	31	91	40	77
≥36 years	5	28	0	0	5	10
Maternal educational level	18	100	32	100	50	100
Incomplete high school	10	56	14	44	24	48
Complete high school or more	8	44	18	56	26	52
Socioeconomic status	18	100	33	100	51	100
A–B1 and B2	1	6	3	9	4	8
C1 and C2	11	61	20	61	31	61
D–E	6	33	10	30	16	31
Trimester of maternal symptoms of Zika Virus infection	15	100	29	100	44	100
First trimester	8	53	21	72	29	66
Second trimester	6	40	7	24	13	30
Third trimester	1	7	1	3	2	5
Maternal concern about development	18	100	34	100	52	100
Yes	15	83	27	79	42	81
No	3	17	7	21	10	19
Maternal concern about behavior	18	100	34	100	52	100
Yes	10	56	14	41	24	46
No	8	44	20	59	28	54
Food insecurity	18	100	34	100	52	100
Yes	5	28	13	38	18	35
No	13	72	21	62	34	65
Risk for maternal depression	18	100	34	100	52	100
Yes	6	33	5	15	11	21
No	12	67	29	85	41	79
Risk for domestic violence	18	100	34	100	52	100
Yes	1	6	3	9	4	8
No	17	94	31	91	48	92

**Table 2 ijerph-19-09554-t002:** Distribution of children (*n* = 52) according to risk of behavioral alteration assessed with the Preschool Pediatric Symptom Checklist of the SWYC questionnaire.

Preschool Pediatric Symptom Checklist to Behavior Disorders						
Variables	At Risk		No Risk		Total	
	*n*	%	*n*	%	*n*	%
Child gender	18	100	34	100	52	100
Male	10	56	18	53	28	54
Female	8	44	16	47	24	46
Prematurity (under 37 weeks)	18	100	34	100	52	100
No	14	78	27	79	41	79
Yes	4	22	7	21	11	21
Head circumference at birth (z-score HC < −2)	18	100	34	100	52	100
Normocephalic (z-score HC ≥ −2)	8	44	3	9	11	21
Microcephalic (z-score HC < −2)	10	56	31	91	41	79
Low birth weight (z-score weight < −2)	18	100	34	100	52	100
No	12	67	29	85	41	79
Yes	6	33	5	15	11	21
Short height at birth (z-score height < −2)	18	100	33	100	51	100
No	13	72	20	61	33	65
Yes	5	28	13	39	18	35
Exclusive breastfeeding up to 6 months	18	100	34	100	52	100
No	9	50	24	71	33	64
Yes	9	50	10	29	19	37
Convulsion	18	100	34	100	52	100
No	7	39	10	29	17	33
Yes	11	61	24	71	35	67
Arthrogryposis	18	100	34	100	52	100
No	15	83	30	88	45	86
Yes	3	17	4	12	7	14
Gastrostomy	18	100	34	100	52	100
No	16	89	31	91	47	90
Yes	2	11	3	9	5	10
Altered ocularfundus	17	100	34	100	51	100
No	9	53	24	71	33	65
Yes	8	47	10	29	18	35
Altered OAE or BERAor both	18	100	33	100	51	100
No	16	89	14	42	30	59
Yes	2	11	19	58	21	41
Strabismus at 2 years	18	100	34	100	52	100
No	9	50	6	18	15	29
Yes	9	50	28	82	37	71
Hospitalization until 36 months	18	100	34	100	52	100
No	10	56	12	35	22	42
Yes	8	44	22	65	30	58
Underweight at 36 months (z-score weight < −2)	11	100	18	100	29	100
No	6	54	7	39	13	45
Yes	5	46	11	61	16	55
Short stature at 36 months (z-score stature < −2)	11	100	18	100	29	100
No	5	46	6	33	11	38
Yes	6	54	12	67	18	62
Child picks up objects	18	100	34	100	52	100
No	9	50	31	91	40	77
Yes	9	50	3	9	12	23
Child sits alone	18	100	34	100	52	100
No	11	61	32	94	43	83
Yes	7	39	2	6	9	17
GMFCS classification	18	100	34	100	52	100
Level I	1	6	0	0	1	2
Level IV	2	11	2	6	4	8
Level V	15	83	32	94	47	90
Child speaks some words	18	100	34	100	52	100
No	10	56	27	79	37	71
Yes	8	44	7	21	15	29

OAE—Otoacoustic Emissions; BERA—Brainstem Evoked Response Audiometry; GMFCS—Gross Motor Function Classification System. N = 51 for initial and final multivariate models. HC—Head Perimeter; OEA—Otoacoustic Emissions; BERA—Brainstem Evoked Response Audiometry; OR—Odds Ratio.

**Table 3 ijerph-19-09554-t003:** Family and child characteristics according to risk of behavior disorder.

Preschool Pediatric Symptom Checklist to Behavioral Disorders				
	N (%)	At Risk	No Risk	*p* Value
Maternal concern about development	42/52 (81)	15/18 (83)	27/34 (79)	1.000 ^a^
Maternal concern about behavior	24/52 (46)	10/18 (56)	14/34 (41)	0.322
Food insecurity	18/52 (35)	5/18 (28)	13/34 (38)	0.451
Risk for maternal depression	11/52 (21)	6/18 (33)	5/34 (15)	0.159 ^a^
Risk for domestic violence	4/52 (8)	1/18 (6)	3/34 (9)	1.000 ^a^
Child picks up objects	12/52 (23)	9/18 (50)	3/34 (9)	0.002 ^a^
Child sits alone	9/52 (17)	7/18 (39)	2/34 (6)	0.005 ^a^
Child speaks some words	15/52 (29)	8/18 (44)	7/34 (21)	0.071
GMFCS classification				0.283 ^a^
Level I	1/52 (2)	1/18 (6)	0/34 (0)	
Level IV	4/52 (8)	2/18 (11)	2/34 (6)	
Level V	47/52 (90)	15/18 (83)	32/34 (94)	
Mean head perimeter at 36 months (cm)	52	11/18 (−4.45)	17/34 (−5.64)	0.1032 ^b^

GMFCS—Gross Motor Function Classification System. ^a^ Chi-Square or Fisher exact test; ^b^ Unpaired *t*-test (two-tailed).

**Table 4 ijerph-19-09554-t004:** Logistic regression models for risk of behavior disorders.

	Univariate Model		Initial Multivariate Model		Final Multivariate Model	
	Crude OR (95%CI)	*p*	Adjusted OR (95%CI)	*p*	Adjusted OR (95%CI)	*p*
Female	0.91 (0.30 to 2.79)	0.866				
Mother’s age (years)	1.1 (0.99 to 1.19)	0.076				
Children’s age (years)	1.03 (0.94 to 1.13)	0.570	1.18 (0.95 to 1.48)	0.140	1.07 (0.91 to 1.26)	0.419
Maternal age at delivery (ref = 19 to 35 years)		0.019		0.067		
≤18 years	4.26 (0.88 to 20.57)	0.071	21.75 (0.91 to 522.16)	0.058		
≥36 years	36.47 (1.84 to 721.38)	0.018	5.74 (0.16 to 209.09)	0.341		
Educational level of the mother—High school	1.58 (0.51 to 4.91)	0.432				
Socioeconomic status (ref. = C)		0.944				
A B	0.76 (0.10 to 5.90)	0.796				
D E	1.10 (0.33 to 3.72)	0.874				
Prematurity	1.14 0.30 to 4.31)	0.849				
Trimester of maternal symptoms(ref. = 1)		0.438				
2	2.19 (0.59 to 8.18)	0.243				
3	2.53 (0.23 to 27.84)	0.448				
Low birth weight (z weight < −2)	2.79 (0.75 to 10.38)	0.126				
Short height at birth (z height < −2)	0.62 (0.19 to 2.07)	0.435				
HC at birth—Normocephalic (ref. = microcephalic)	7.29 (1.75 to 30.36)	0.006	40.02 (3.17 to 505.5)	0.004	35.84 (3.49 to 367.9)	0.003
Low weight at 36 months (z weight < −2)	0.55 (0.13 to 2.38)	0.425				
Short height at 36 months (z height < −2)	0.61 (0.14 to 2.70)	0.519				
Arthrogryposis	0.53 (0.33 to 7.03)	0.584				
Altered ocularfundus	2.9 (0.65 to 6.76)	0.219				
Altered OAE or BERA or both	0.11 (0.03 to 0.50)	0.004	0.13 (0.02 to 1.04)	0.054	0.07 (0.01 to 0.49)	0.007
Convulsions	0.66 (0.20 to 2.12)	0.482				
Strabismus at 2 years	0.23 (0.07 to 0.79)	0.019	0.06 (0.01 to 1.06)	0.055	0.05 (0.01 to 0.36)	0.003
Constipation	0.36 (0.08 to 1.65)	0.188				
Hospitalization	0.45 (0.14 to 1.40)	0.169				
Exclusive breastfeeding up to 6 months	2.33 (0.74 to 7.40)	0.150				
Gastrostomy	1.36 (0.24 to 7.68)	0.725				

N = 51 for initial and final multivariate models. HC—Head Perimeter; OEA—Otoacoustic Emissions; BERA—Brainstem Evoked Response Audiometry; OR—Odds Ratio.

## Data Availability

De-identified data are available after a personal email request to the authors Nívia Arrais (niviaarrais@gmail.com) and Silvana Pereira (silvana.alves@ufrn.br).

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
