# Peer review of "Factors Associated with Behavioral Disorders in Children with Congenital Zika Syndrome and Their Families—A Cross-Sectional Study"

_ijerph, 2022, doi:10.3390/ijerph19159554_

Round 1

Reviewer 1 Report

This is an interesting piece of research that aims to characterise the epidemiological, clinical and familial context of children with CZS and to identify factors associated with the risk of behavioural disorders.  I have the following comments to make;

1. The introduction is very brief and there is insufficient rationale about why the study is needed or why the specific variables are of interest.  The case for the study needs to be made more comprehensively and clearly by expanding on what is already written.

2. Sample size - A total of 52 participants were included in this study.  This does not appear to be sufficiently powered to run a logistic regression where several hundred a more likely needed.  Can the authors justify why this statistical test was run and how they estimated the sample size needed?

3. Sample size - there is a very brief mention of sample size as a study limitation in the Discussion section.  More needs to be said to explain why the sample size is a limitation, what the implications are, and what this means when other researchers are interpreting their findings.

4. Data were collected at assessments when the infant was between 2-5 years of age.  This appears to be a large range and introduces the potential for confounding factors to explain your findings.  What inclusion/exclusion criteria were implemented to ensure the data collected was reliable and consistent.  Is it possible that the lack of a specific data collection point (e.g. when children were at a set age that could have been consistently applied across your sample) might impact your results?  Did you attempt to control for the age of child when running statistical tests? A more detailed explanation of your study's inclusion/exclusion criteria is needed and a more in-depth evaluation is needed too.

Author Response

Thank you very much for your suggestions. We have carefully considered your comments and revised the paper based on those comments and recommendations. Please find the answers to your comments attached.

Reviewer 2 Report

The work is very relevant to the subject itself, even though a decrease in cases has been observed, but epidemiologically, new waves of new cases may come. Arboviruses are endemic mainly in developing countries. 

1. In line 79 in relation to the definition of microcephaly, there should be a greater specification as observed in the article by: Cad Saude Publica . 2021 Nov 22;37(11):e00228520. doi: 10.1590/0102-311X00228520. 

2.Table 1 is too big. difficult in my view to read and follow. I suggest splitting between maternal and child data.

Author Response

(The authors gave the same response as above.)

Round 2

Reviewer 1 Report

Thank you very much for your response to my points.  I feel that everything has been adequately addressed.  I look forward to seeing the paper in print!